# Characterization of flow recirculation zones at the Perdigão site using multi-lidar measurements

Robert Menke[1], Nikola Vasiljević[1], Jakob Mann[1], and Julie K. Lundquist[2,3]

[1]Technical University of Denmark - DTU Wind Energy, Fredriksborgvej 399, Building 118, 4000 Roskilde, Denmark
[2]Department of Atmospheric and Oceanic Sciences, University of Colorado, Boulder, Colorado, USA
[3]National Renewable Energy Laboratory, Golden, Colorado, USA

**Correspondence:** Robert Menke (rmen@dtu.dk)

**Abstract.** Because flow recirculation can generate significant amounts of turbulence, it can impact the success of wind energy projects. This study uses unique Doppler lidar observations to quantify occurrences of flow recirculation on lee sides of ridges. An extensive dataset of observations of flow over complex terrain is available from the Perdigão-2017 field campaign over a period of three months. The campaign site was selected because of the unique terrain feature of two nearly parallel ridges with
5  a valley-to-ridge-top height difference of about 200 m and a ridge-to-ridge distance of 1.4 km.

Six scanning Doppler lidars probed the flow field in several vertical planes orthogonal to the ridges using range-height-indicator scans. With this lidar setup, we achieved vertical scans of the recirculation zone at three positions along two parallel ridges. We construct a method to identify flow recirculation zones in the scans, as well as define characteristics of these zones. According to our data analysis, flow recirculation, with reverse flow wind speeds greater than $0.5\,\mathrm{m\,s^{-1}}$, occurs over $50\%$ of
10  the time when the wind direction is perpendicular to the direction of the ridges. Atmospheric conditions, such as atmospheric stability and wind speed, affect the occurrence of flow recirculation. Flow recirculation occurs more frequently during periods with wind speeds above $8\,\mathrm{m\,s^{-1}}$. Recirculation within the valley affects the mean wind and turbulence fields at turbine heights on the downwind ridge in magnitudes significant for wind resource assessment.

## 1 Introduction

15  Traditional wind turbine siting relies on wind measurements from a single mast or a small number of masts deployed at the site of interest. These measurements are extrapolated over the entire site using a linearized flow model to provide a general assessment of wind resources (Troen and Petersen, 1989). This method proved suitable for flat terrain sites, gently sloping terrains and offshore where the flow is close to being homogeneous. In complex terrain, however, where wind fields are strongly affected by the topography and roughness changes, the uncertainties of wind resource estimations are increased because models
20  struggle to predict flow under these conditions, despite of continues advancement of flow models. A vast number of flow phenomena occur at sites with complex geometry (Rotach and Zardi, 2007), such as flow acceleration and channeling effects, the formation of lee waves, and flow recirculation (Stull, 2012) which are not captured well by frequently applied computer models (e.g. the Wind Atlas Analysis and Application Program, WASP) in the wind farm planning process. These effects have high spatial and temporal variability and are thus difficult to investigate with sparse measurements. However, these phenomena

can induce high loads on wind turbines which will decrease turbine lifetime (Sathe et al., 2013), and therefore such effects must be characterized and understood in order to be able to predict them.

In this study, we investigate flow recirculation occurring in the lee of ridges. Flow recirculation can be identified by flow moving reversely to the prevailing flow direction. In this region both turbulence and wind shear are increased (Stull, 2012). Recirculation occurs behind abrupt drops in terrain elevation such as cliffs, but also in the lee of gentle slopes of a certain steepness (Wood, 1995; Xu and Yi, 2013; Kutter et al., 2017). In addition, small terrain features (Lange et al., 2017) and the presence of forest canopy (Sogachev et al., 2004; Finnigan and Belcher, 2004) can modify the occurrence of recirculation drastically. Furthermore, the stratification of the atmosphere influences the occurrence of recirculation; for instance, recirculation was prevalent during neutral and unstable atmospheric conditions during the observational study of Kutter et al. (2017). The length of flow separation zones is mainly dependent on the hill shape and the surface roughness. Increasing downwind slopes and larger roughness is causing longer separation zones. In Kaimal and Finnigan (1994) a usual extend of the separation zones for naturally shaped hills of two to three times the hill height is stated.

Many studies that are describing recirculation or in general flow over complex terrain rely on experimental data measured in the field. Probably the best known field experiment took place at the Askervein hill (Taylor and Teunissen, 1983). The measured data is heavily used to validate flow simulations such as linear models but also more sophisticated code using Reynolds averaged Navier-Stokes (RANS) methods (Silva Lopes et al., 2007) and large-eddy simulations (LES) as presented in Chow and Street (2009). Another, more recent field experiment is the Bolund hill campaign which took place in 2007-2008. A blind test of different microscale flow models for the Bolund case revealed major uncertainties in predicting the influence of the Bolund escarpment on the flow field. The experiment also demonstrated the strength of lidars, in their case continuous-wave Doppler lidars, to measure the influence of the Bolund escarpment on the flow field (Lange et al., 2016). Here, we investigate measurements from the Perdigão 2017 campaign where, in addition to sonic anemometers on 100-m high masts, pulsed Doppler lidars observed the flow over the two parallel ridges. To surpass the capabilities of point measurements from networks of masts, Doppler lidars have, not only in the Bolund campaign, proved capable of measuring flow phenomena such as wind turbine and wind farm wakes (Käsler et al., 2010; Iungo et al., 2013; Aitken et al., 2014; Menke et al., 2018) and the effect of roughness changes on wind fields in different heights and nearshore flows (Mann et al., 2017). In addition to the advancements in the measurement technology, the orography of the Perdigão site can be approximated by two two-dimensional ridges as the Askervein's orography with an three-dimensional hill of about half the height of the Perdigão ridges.

The present work uses multi-lidar measurements to observe and classify recirculation that occurs in the lee of ridges at Perdigão. The flow field is measured by six scanning lidars at three different transects across the ridges. The scans cover the entire recirculation zone, enabling us to characterize the dimensions of recirculation zones and to investigate their spatial variability along the ridges. Moreover, we relate these recirculation zones to resulting changes in turbulence and wind shear measured by sonic anemometers on measurement masts at potential wind turbine sites downwind of recirculation zones. The measurement campaign, Perdigão 2017 (Mann et al., 2017; Fernando et al., 2018) is presented in Section 2 of this manuscript. In Section 3, we describe how the lidar measurements are used to detect recirculation zones. The occurrences of recirculation

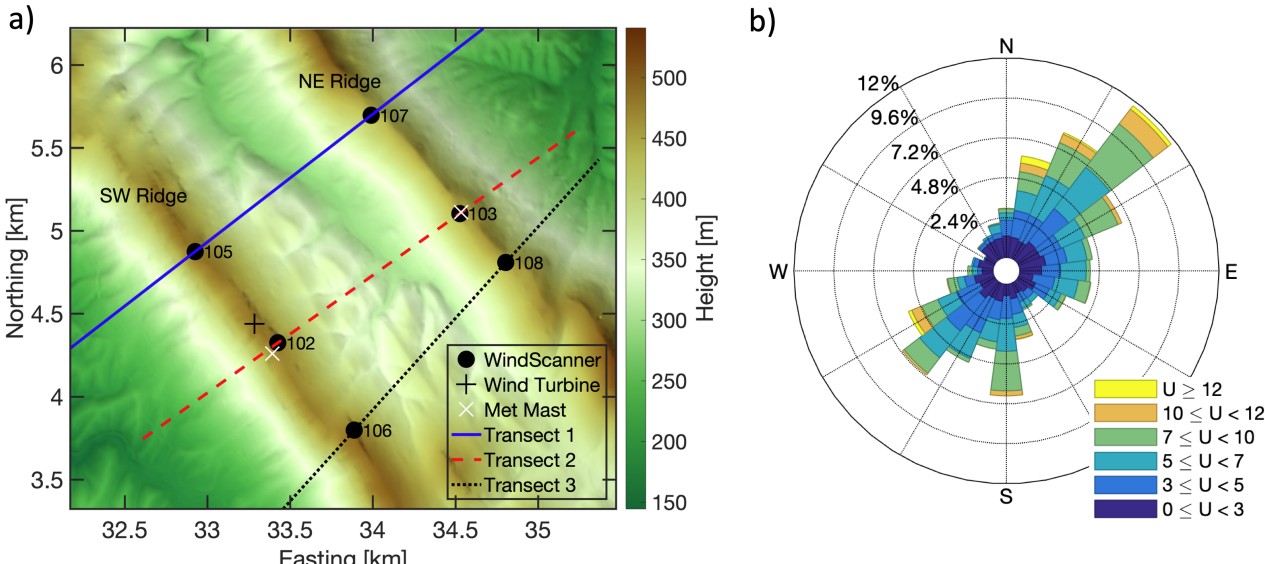

**Figure 1.** a) Elevation map of the Perdigão site. Measurement transects are indicated by the solid blue (Transect 1), dashed red (Transect 2) and dotted black (Transect 3) lines. Black disks: WindScanner positions; Numbers: WindScanner station number as in Table 1; Black +: wind turbine position; White X's: position of measurement masts used in this investigation. PT-TM06/ETRS89 coordinate system, height above sea level. b) Distribution of 30-minute averaged wind measurements taken by the 100 m anemometer on the mast located on the northeast ridge over the period from April 1 to June 15, 2017. Measurements in $\mathrm{m\,s^{-1}}$.

at the Perdigão site are analyzed with respect to mean wind speed and atmospheric stability in Section 4. In addition, we assess the impact of recirculation at potential turbine sites in this section. Conclusions are given in the last section.

## 2  Field campaign overview and measurement data

The experiment is part of a series of measurement campaigns within the New European Wind Atlas with the objective to provide
5   an extensive dataset of high-quality measurements of flow over complex terrain (Mann et al., 2017). Starting in January 2017, a network of 50 measurement masts was deployed at the site, and over the next few months, 19 scanning Doppler lidars were added. For this study, we use the data from two 100-meter masts equipped with 3D ultrasonic anemometers (Gill WindMaster Pro) and temperature sensors (NCAR SHT75) located on the tops of the ridges as well as six scanning lidars (Figure 1 and, for exact positions, Table 1). Here, we focus on the period of 68 days from April 8 to June 14 during which long-range
10   WindScanners and masts operated in parallel.

**Table 1.** Positions of instruments used in this study. Easting and northing in PT-TM06 / ETRS89 coordinate system and elevation above sea level. Elevation of the masts (first two rows) refers to the mast base and elevation of the WindScanners (remaining rows) refers to the scanner head position.

| Station Number | Station Acronym | Easting [m] | Northing [m] | Elevation [m] |
|:---:|:---:|:---:|:---:|:---:|
| 20 | 20/tse04 | 33392.7 | 4259.6 | 473.0 |
| 29 | 29/tse13 | 34533.6 | 5112.0 | 452.9 |
| 105 | LRWS #5(DTU) | 32926.5 | 4874.3 | 485.9 |
| 107 | LRWS #7(DTU) | 33990.6 | 5695.3 | 437.1 |
| 102 | LRWS #2(DTU) | 33426.2 | 4324.1 | 480.3 |
| 103 | LRWS #3(DTU) | 34526.3 | 5103.5 | 452.3 |
| 106 | LRWS #6(DTU) | 33888.7 | 3798.0 | 486.3 |
| 108 | LRWS #8(DTU) | 34804.6 | 4807.9 | 452.8 |

## 2.1 Measurement site

The measurement site, located in central Portugal near the Spanish border, was selected because of the unique terrain feature of two nearly parallel ridges with a valley-to-ridge-top elevation difference of about 200 m and a ridge-to-ridge distance of 1.4 km (Figure 1). The ridges run from the northwest to the southeast and are hereafter referred to as SW and NE ridge, respectively. The terrain roughness is characterized by a patchy vegetation of pine and eucalyptus trees with heights of up to 15 m. A detailed site description can be found in Vasiljević et al. (2017).

## 2.2 WindScanner measurements

We deployed six long-range WindScanners, scanning-Doppler lidars (Vasiljevic et al., 2016), on top of the ridges at the Perdigão site. All scanners performed range-height-indicator (RHI) scans along three transects which are almost perpendicular to the ridges. The transects are orientated along the SW-NE direction (Figure 1, see Table 2 for the exact directions). The scans with a maximal range of 3 km covered the area within the valley. Outside the valley, the scans covered areas to a maximum range of 3 km and up to 1600 m above ground level. Elevation profiles along the transects are shown in Figure 2. We measured continuously during the upwards movement of the scanners with an averaging elevation angle of 0.75° over a range of 25°. Range gates (time intervals used for determining the wind speed from the back-scattered light. These time intervals corresponds to spatial intervals along the line-of-sight (LOS) for which the LOS wind speed is evaluated. They translate into a weighting function along the LOS which in this case has a full-width half-maximum of approximately 30 m.) were placed every 15 m, starting at a range of 100 m extending to 3000 m. The effective probe length of the scans was 35 m along the laser light propagation path. The RHI scans were carried out twice per hour for 10 minutes by all lidars. During one 10-minute period, 23 scans were performed; a single scan took about 26 seconds to complete. We averaged the data over each 10-minute period, after filtering the data by the carrier-to-noise-ratio (CNR $< -24$ dB) and removing hard target influences. To ensure that

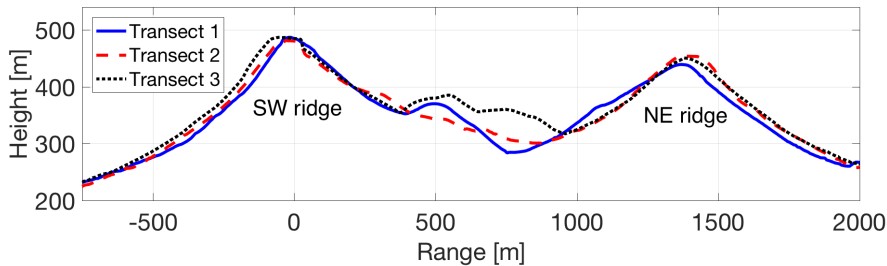

**Figure 2.** Elevation profiles of measurement transects. The transect colors correspond to Figure 1. The vertical scale is exaggerated by a factor of two. The origin (range = 0) refers to the position of WindScanner 105, 102 and 106 for transect 1, 2 and 3, respectively. The coordinates of the transects are available as CSV files in the supplementary material.

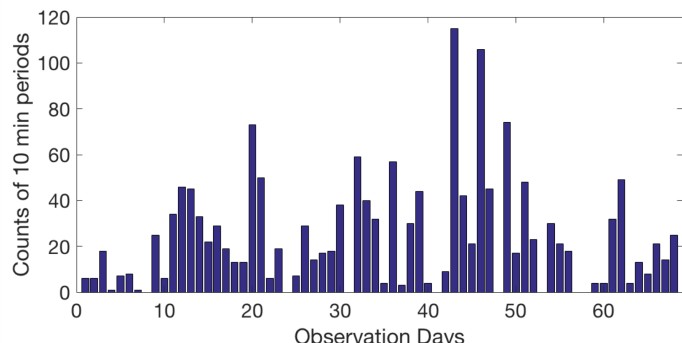

**Figure 3.** Distribution of the 1613 available 10-minute periods with wind directions perpendicular to the ridges (214° – 254° and 34° – 74°). The observational period lasted for 68 days starting on 2017/04/08 and ending on 2017/06/14. The intensive observation period of the campaign started on day 24 (2017/05/01). The availability data can be found the in the supplementary material.

the pointing accuracy of minimum 0.05° is maintained during the entire measurement period, the WindScanner's leveling was checked and the known hard targets were mapped consistently through the measurement campaign (Vasiljević et al., 2017).

For the analysis, we selected periods within two 40° wind direction sectors centered at a line perpendicular to the ridge (214° – 254° and 34° – 74°). The direction is measured by the 100-m sonic anemometer on the mast located on the southwest ridge. In total, 1619 10-minute periods of measurements are available for the analysis in these sectors (Figure 3).

## 3 Methods

### 3.1 Atmospheric stability

To define atmospheric stability, we calculated the gradient Richardson number at the mast located on the Northeast ridge (Figure 1). The SW ridge mast could not be used due missing temperature measurements for a major part of the observational

**Table 2.** Parameters describing the measurement transects. Peak height gives the highest points at the southwest (SW) and northeast (NE) ridge, respectively. The average terrain elevation is measured along the transects in between the peaks. The maximum height of the ridges relative to the terrain height at the transect before and after the ridges is given by $h_{SW}$ and $h_{NE}$. $A_{SW}$ and $A_{NE}$ is the half-width at half-height of the faces inside the valley.

| Transect | Orientation [deg] | Peak height SW ridge [m] | Peak height NE ridge [m] | Average height [m] | Lowest point in valley [m] | Peak-to-peak distance [m] | $h_{SW}$ [m] | $A_{SW}$ [m] | $h_{NE}$ [m] | $A_{NE}$ [m] |
|---|---|---|---|---|---|---|---|---|---|---|
| 1 | 52.3 | 487.1 | 439.3 | 369.7 | 283.5 | 1388.0 | 278.6 | 601.6 | 230.8 | 402.2 |
| 2 | 54.7 | 481.6 | 453.8 | 369.2 | 300.3 | 1425.6 | 268.4 | 488.6 | 240.5 | 345.8 |
| 3 | 42.2 | 487.0 | 450.7 | 384.2 | 317.2 | 1435.8 | 237.3 | 383.3 | 201.1 | 278.8 |

periods. The gradient Richardson number is defined as in Stull (2012) :

$$Ri_G = \frac{\frac{g}{\Theta} \frac{\partial \overline{\Theta}}{\partial z}}{\left[ \left( \frac{\partial \overline{u}}{\partial z} \right)^2 + \left( \frac{\partial \overline{v}}{\partial z} \right)^2 \right]} \tag{1}$$

where $\Theta$ is the mean temperature measured at $100\,\mathrm{m}$, $\partial \overline{\Theta}$ the potential temperature difference between $10\,\mathrm{m}$ and $100\,\mathrm{m}$, $g$ = $9.81\,\mathrm{m\,s^{-2}}$ is the gravitational acceleration, and $u$ and $v$ are the two horizontal components of the mean wind vector. The potential temperature is approximated by $\overline{\Theta} \approx T + (g/C_p) \cdot z$ where $g/C_p = 0.0098\,\mathrm{K m^{-1}}$ (Stull, 2012). In order to reduce the effect of terrain-induced flow on the stability estimates, we calculate the difference between the wind speed measured by the $100\,\mathrm{m}$ sonic and at ground level, $0\,\mathrm{m}$, where the wind speed is assumed to be zero. A similar approach has been used by Burns et al. (2011). All calculations are based on 30-minute averages. In total we could assess the atmospheric stability for 82% of the 10-minute periods due to missing temperature data from the meteorological mast.

Moreover, the Brunt–Väisälä frequency which is defined as:

$$N = \sqrt{\frac{g}{\Theta} \frac{\partial \overline{\Theta}}{\partial \overline{z}}} \tag{2}$$

is estimated from the mast measurements for stably stratified conditions. We use again the potential temperature difference between the $10\,\mathrm{m}$ and $100\,\mathrm{m}$ levels at the mast on the northeast ridge.

### 3.2 Detection of recirculation zones

We develop a method to identify recirculation zones in measurements taken by lidars along the measurement transects (Figure 5). This method is based on measurements by two lidars located on the same transect. By using two lidars, the area covered by the scans within the valley can be maximized. The RHI scans provide LOS wind speed measurements in a polar coordinate system with the lidar at the origin. Consequently, two lidars which are measuring in the same vertical plane will provide data in polar coordinate systems with different origins. Therefore, the LOS wind speeds measured by the lidars are interpolated to a Cartesian coordinate system with the abscissa orientated along the measurement transect and a grid size of $10\,\mathrm{m}$ in both the

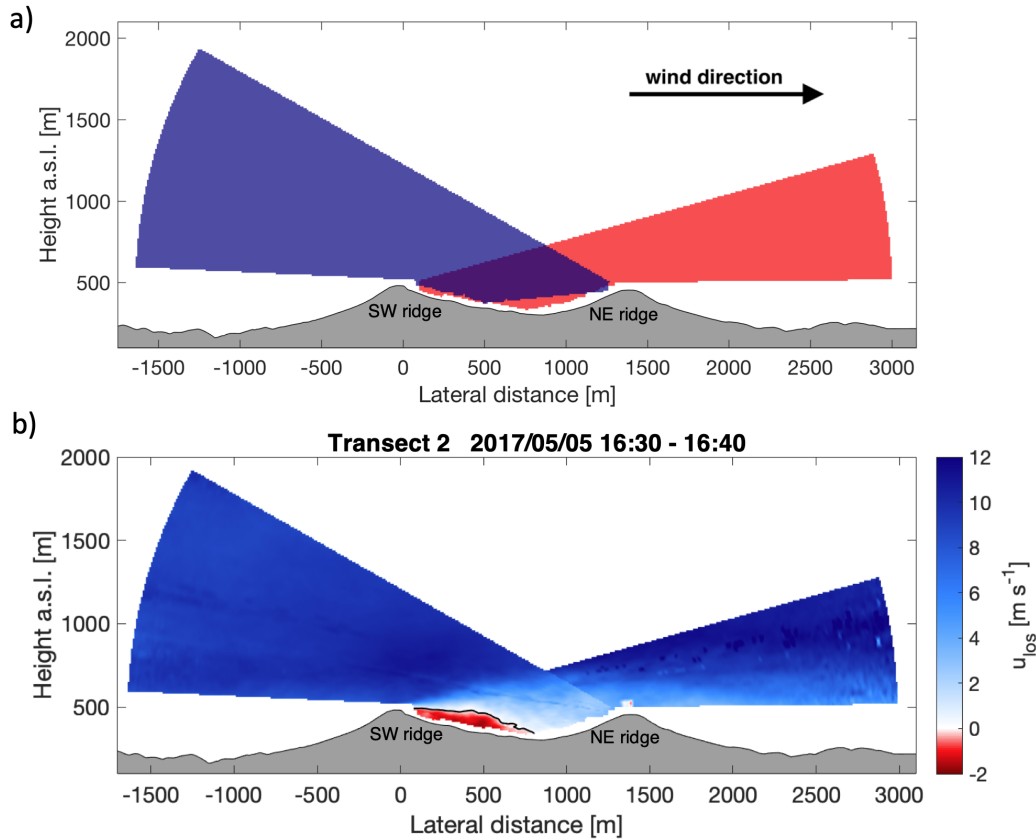

**Figure 4.** Detection of recirculation zones at the second transect (lateral distance = 0 is the position of WindScanner 102), view from the southeast. a) Schematic of the overlay of the two RHI scans at the transect. Blue and red areas are corresponding to the RHI scans and the grey area shows the terrain. b) Lidar measurements along the transect and detected recirculation zone for one 10-min period with $\text{Ri}_G =$ 0.016. Positive LOS velocities indicate flow towards the northeast (from left to right in the plot). The terrain is plotted in grey and the contour line defining the recirculation zone is plotted in black inside the velocity field. Time in UTC.

vertical and horizontal directions. In overlap regions, only measurements of the lidar further downwind are used. Additional regions that are sampled by the second lidar upwind are filled in a second step. This procedure prevents discontinuities between the two scans from interfering with estimates within the recirculation zone (Figure 4a). We do not attempt to combine LOS velocities measured by the lidars to get the true horizontal wind along the measurement transect, since the elevations angles are too small to measure the vertical component precisely. Moreover, the assumption of zero vertical wind speed, as often applied in flat terrain.

Having the measurements combined, the zero contour line of the velocity field behind the upwind ridge is determined using linear interpolation. The area bounded by this contour and the terrain beneath describes the dimensions of the recirculation zone (Figure 4). Recirculation is defined to be presented for reverse flow speeds greater than $0.5\,\text{m}\,\text{s}^{-1}$. Also, the length of

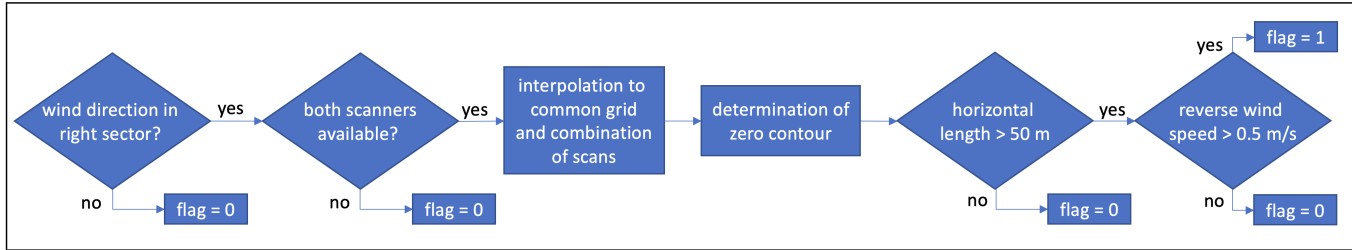

**Figure 5.** Flow diagram of recirculation zone detection process. Recirculation is detected for periods that are flagged with "1", no recirculation is detected for periods flagged with "0".

these zones is required to exceed five grid cells or 50 m in the horizontal dimension. To increase the robustness of this method we require that at least one valid measurement is available below the contour line. A potential disadvantage is that recirculation appearing close to the ground cannot be captured since measurement range gates of the lidars that reach into the ground or vegetation must be discarded due to the hard target returns.

## 4 Results and Discussion

The method to detect the recirculation zones is applied to all available periods so that we can analyze the occurrences of recirculation under different atmospheric conditions in combination with the mast measurements. Specifically, we assess the impact of flow direction, mean flow speed and atmospheric stability. The measurement period from April to mid of June was generally very hot with maximum temperatures above 40°C. Days were normally cloud-free while on some mornings fog formed inside the valley. Therefore, stably stratified atmospheric conditions at night and unstable conditions during the day are prevailing.

### 4.1 Occurrence of recirculation

On average, recirculation occurs frequently, in 52% of all 10-min time periods examined. This ratio changes for the individual transects. Along transect 1 (the furthest northwest, with the lowest point in the valley and the shortest distance between ridges), recirculation occurs most frequently at 69% of the time, while transect 2 observes recirculation 51% of the time. Transect 3 (the furthest southeast, with the highest low point in the valley and the longest peak-to-peak distance) only observes recirculation in 32% of the available periods. These variations of recirculation occurrence may be related to the transects' elevation profiles within the valley. The average elevation along transect 1 and 2 is the same, whereas the average elevation along transect 3 is 15 m higher (Table 2). Additionally, the elevation profile of transect 3 shows a hill at the center of the valley (Figure 2). We assume that these features suppress the formation of recirculation along transect 3. Other terrain parameters and land use/land cover characteristics only vary insignificantly among the transects.

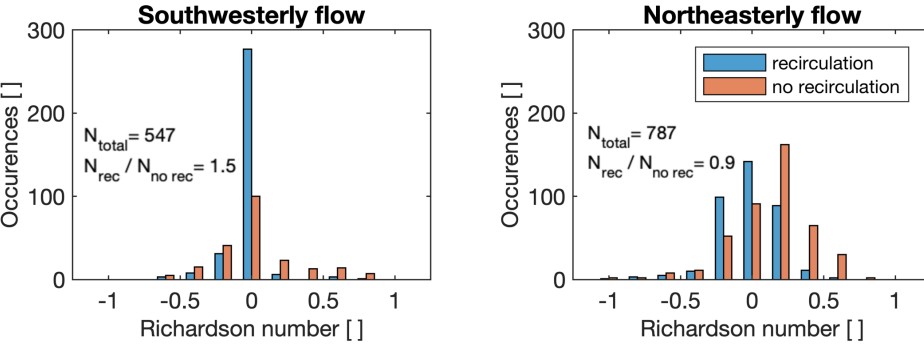

**Figure 6.** Recirculation occurrence as a function of the Richardson number. The results are presented in bins with a width of 0.2. Data from all three transects are presented together, segregated by the wind direction.

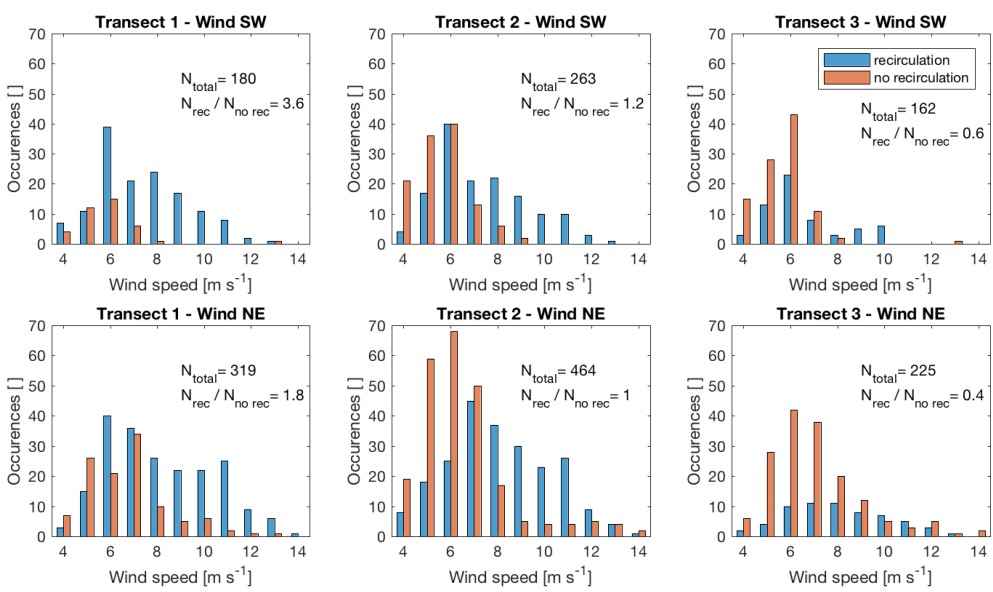

**Figure 7.** Occurrence of recirculation depending on the mean wind speed measured by the 100 m sonic on the upstream mast. The histograms are separated by transect and wind direction.

### 4.1.1 Dependence on atmospheric stability

Recirculation is more likely to occur during unstable or neutral atmospheric conditions ($Ri \leq 0$) than for stable conditions ($Ri > 0$) for both SW and NE winds (Figure 6), as expect for flow over steep geometries such as the Perdigão ridges. These results resonate with the findings of Kutter et al. (2017), who found, in a study of recirculation at a single forested hill, that

recirculation is less likely to occur under stable conditions. In fact, they found only one period with recirculation during stable conditions, whereas we have several such cases of recirculation under stable conditions.

Only the bin centered at $Ri = -0.2$ shows an opposite dominance of recirculation versus no recirculation for the two flow directions. Recirculation occurs in this bin mainly for northeasterly flow (Figure 6b), whereas the opposite is true for southwesterly flow (Figure 6a). For stability calculations, only measurements from the mast on the northeast ridge are used since the availability of temperature measurements at the other mast is very low. Thus, turbulent mixing is increased at the mast location for southwesterly flow which could affect the Richardson number and thereby these results.

### 4.1.2 Dependence on mean wind direction

Recirculation occurs more often for southwesterly wind directions (Figure 7). A possible explanation for this difference could be that the northeast face of the southwest ridge (the downwind face in southwesterly flow) has steep escarpments close to the ridge top and the average slope is slightly steeper compared to the southwest face of the northeast ridge (the downwind face in northeasterly flow). Both, the higher steepness and the escarpments, make flow separation more likely. At transect 1 recirculation occurs predominantly, especially for southwesterly flow. The ratio of periods with recirculation occurrence to no occurrence is balanced at transect 2, and at transect 3 recirculation can only be observed 32% of the time. For all transects the chance of recirculation is higher for southwesterly wind directions.

### 4.1.3 Dependence on mean wind speed

More differences between the transects emerge when we group the data by wind speed. Transects 1 and 2 show a very high chance of recirculation to occur for wind speeds above $8\,\mathrm{m\,s^{-1}}$ (Figure 7). For wind speeds above $8\,\mathrm{m\,s^{-1}}$, fewer cases of recirculation occur for northeasterly wind directions, whereas recirculation is present in almost all cases for southwesterly wind directions. Moreover, the wind speed bin of $6\,\mathrm{m\,s^{-1}}$ shows interesting results in the inter-transect comparisons: at transect 1, recirculation dominates this bin, whereas a contrary trend can be observed at the other transects.

### 4.1.4 Characteristics under stratified conditions

The findings above show that recirculation occurs as expected prevailingly under unstable atmospheric conditions but occurrences under stably stratified conditions are not absent. In general, three types of behaviours can be expected when flow encounters an obstacle (e.g. a hill or a ridge), (i) complete attachment of the flow, separation downstream is suppressed by strong stratification; (ii) downstream separation of flow direction directly behind the obstacle; and (iii) post-wave separation (Baines, 1998). The occurrence of these behaviours is sensitive to the downstream slope $h/A$ of the obstacle and $Nh/u$. Steeper downstream slopes support the formation of flow separation behind an obstacle and stronger stratification and higher values (separation occurs when $NA/u < \pi$) of $Nh/u$ suppress separation respectively. The relation of these mechanisms is well summarized in Figure 5.8 in Baines (1998) and we present our findings in the same non-dimensional framework in Figure 9. The observations made in Perdigão, recirculation and non recirculation periods, are falling almost entirely in an area

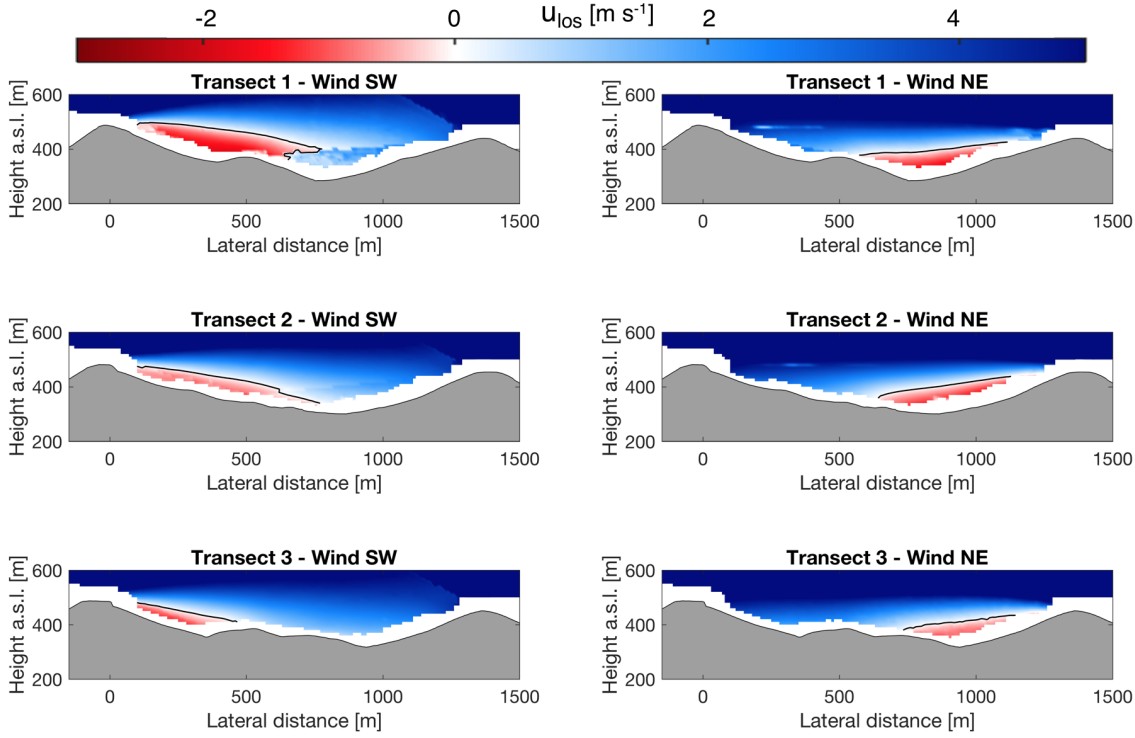

**Figure 8.** Zero contour line (black line) defining the upper border of recirculation zones of the mean velocity field of all recirculation periods per transect and flow direction. The flow speed is positive in the direction of the mean flow field and the transect origins (lateral distance = 0) refers to the position of the WindScanner 105, 102 and 106 for transect 1, 2, and 3, respectively.

where $NA/u < \pi$ in which separation is expected. However, looking at the mean value for recirculation and non recirculation periods it becomes noticeable that the mean of non recirculation periods of $Nh/u$ is higher consistently at all transects. The higher mean value of non recirculation periods aligns with the expectations described above to find attached flow or post-wave separation (which is not necessarily detected by our dectection method) for higher values of $Nh/u$.

## 4.2 Recirculation zone characteristics

The observed recirculation zones can be described by their shape in terms of extent and height and the magnitude of reverse flow speed. On average, the observed recirculation areas extend to 697 m in the horizontal for both wind directions, which is almost exactly half the average ridge-to-ridge distance at the transects. At the southeastern transect (transect 3, the longest transect with the least frequent recirculation), the average extent is 22% smaller than the mean of all transects (Figure 4). The average extent at the northwestern transect (transect 1, the shortest transect), is 9% higher than the average extent. The average depth of observed recirculation zones is 157 m above the valley bottom height and changes insignificantly among the transects, whereas the maximum depth shows larger variation among the transects. We found that recirculation zones at transect 1 reach

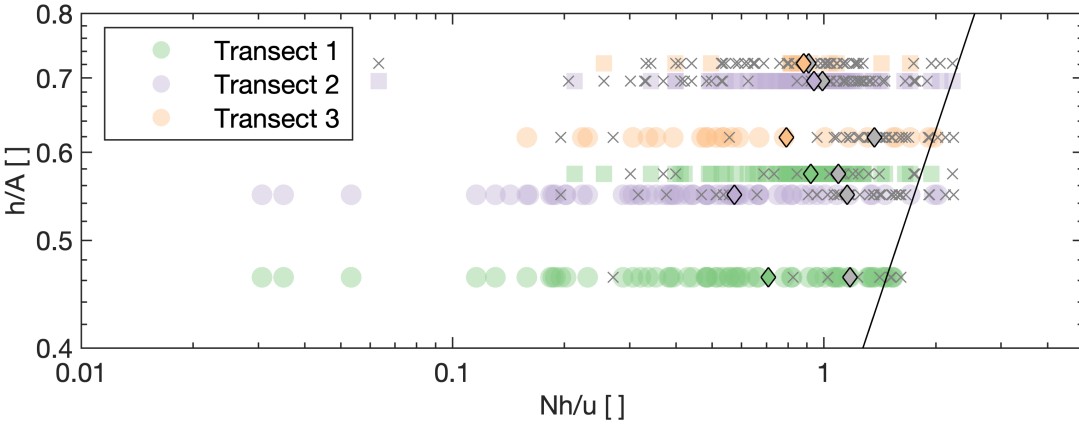

**Figure 9.** Observational periods with recirculation (colored markers) and without recirculation (gray cross marks) under stratified conditions as a function of mean lee-side slope ($h/A$) and $Nh/u$. The round markers refer to recirculation at the southwest ridge and the square markers to northeast ridge. The colored diamond markers show the mean value of $Nh/u$ per transect during periods with recirculation and the gray markers the mean for periods without recirculation. The solid black line shows where $NA/u = \pi$.

up to 300 m above the valley bottom and 215 m and 189 m at transect 2 and 3, respectively. Maximum reverse flow speeds of above $4\,\mathrm{m\,s^{-1}}$ are observed. The median reverse flow speeds at the transects are 2.0, 1.5 and $1.4\,\mathrm{m\,s^{-1}}$ for transect 1, 2 and 3, respectively. For all transects and both flow directions combined, a trend for higher reverse flow speeds for Richardson numbers close to zero is found (Figure 10). For all transects and both flow directions combined the relation of the reverse flow speeds to the Richardson number is presented in Figure 10. The reverse flow speed measurements are divided by the inflow wind speed measured at the meteorological masts. For unstable atmospheric conditions (Ri < 0) no clear trend for the relation of reverse wind speed and inflow wind speed can be observed. Ratios from less than 0.1 to more than 0.5 can be observed. During conditions with Ri > 0 rations are generally lower and appear to decrease with increasing Ri. For high wind speeds (greater than $12\,\mathrm{m\,s^{-1}}$) ratios of less than 0.3 are observed and they occur, as expected, for small or slightly positive Richardson numbers.

### 4.3 Impact on potential turbine sites

Ridges and hills are advantageous sites for wind turbines, due to the terrain-induced speed-up (Hunt and Snyder, 1980). However, these stronger winds can come at a cost, due to higher turbulence levels and gustiness which increases loads and decreases the lifetime of turbines. In this section, we assess differences in wind characteristics at the downwind ridge for both the entire dataset as well as a specific case study of the period shown in Figure 4.

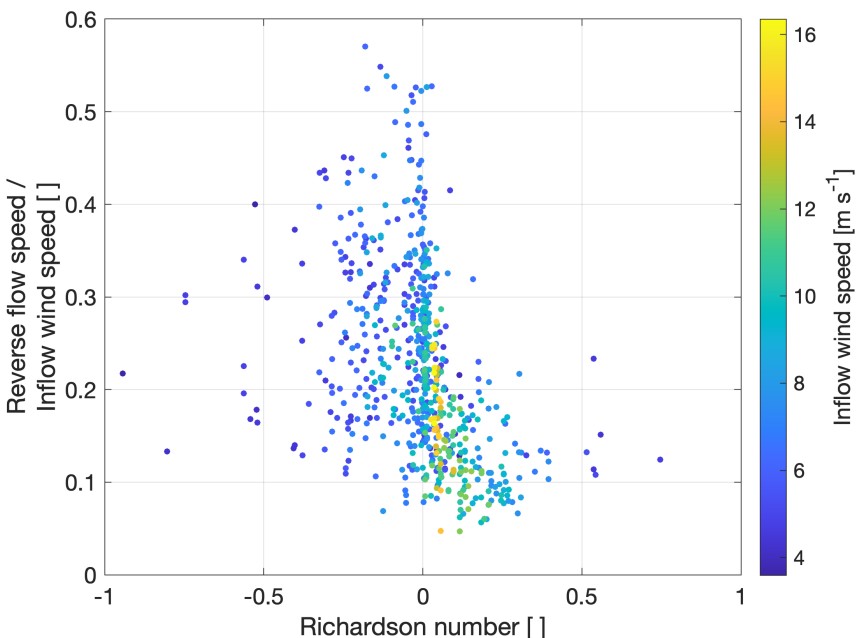

**Figure 10.** Maximum reverse flow speed divided by the inflow wind speed over the Richardson number. The color scaling shows the wind speed measured at the upstream mast 100 m above ground.

### 4.3.1 Case study of recirculation under neutral stratification

On May 5 from 16:30 to 16:40 UTC, winds were $10.4\,\mathrm{m\,s^{-1}}$ from the southwesterly direction (254°). A strong and distinct recirculation zone occurred behind the southwest ridge within the valley at the second transect (Figure 4). The zone has a total length of 807 m, spreading out beyond the valley center. It reaches over the ridge-peak height to 192 m above the valley bottom.

5    Reverse flows inside the recirculation zone have a magnitude of up to $1.8\,\mathrm{m\,s^{-1}}$, or 17% of the inflow.

For this case with a strong recirculation zone we observe a reduction in mean wind speed and increase the turbulence intensity at the northeast ridge. Measurements at the meteorological masts show the impact at the downwind ridge (Table 3). At the 100-m level, the wind speed is reduced 50% as compared to the upwind 100-m mast. Similarly, the turbulence intensity increases from 12% at the upwind ridge to 31% at the downwind ridge. The wind veer observed from 60 to 100 m equals 14°

10    at the downwind ridge versus 2° at the upwind. Moreover, the lidar scan shows that the wake of the upwind ridge reaches to approximately 250 m above the northeast peak height (Figure 4).

### 4.3.2 General impact of recirculation at downwind locations

The detailed analysis of the period in the previous section shows significant changes of the wind characteristics at the downwind ridge. To generalize this observation, we analyze all periods available from transect 2 (on which the measurement masts

**Table 3.** Comparison of wind characteristics during the period 2017/05/05 16:30 - 16:40 UTC measured by the sonic anemometers at 100-m height on the upwind southwest (SW) and downwind northeast (NE) ridges. Turbulence intensity is defined as $I = \sigma_U \overline{U}^{-1}$ where $\overline{U}$ is the mean wind speed and $\sigma_U$ the standard deviation of $U$. Turbulent kinetic energy is calculated as $e = \frac{1}{2}\left[\overline{u'^2} + \overline{v'^2} + \overline{w'^2}\right]$, where $u'$, $v'$ and $w'$ are the fluctuating parts of the wind vector components over a 10-min average as measured by the sonic anemometers.

| | SW ridge (100m) | NE ridge (100m) |
|---|---|---|
| Wind speed, $\overline{U}$ [m s$^{-1}$] | 10.4 | 5.2 |
| Wind direction [°] | 254.1 | 239.7 |
| Turbulence intensity, $I$ [%] | 11.8 | 30.7 |
| Turbulent kinetic energy, $e$ [m$^2$s$^{-2}$] | 2.26 | 4.32 |
| Wind veer (60-100 m) [°] | $-2.1$ | $-13.7$ |

**Table 4.** Changes in the wind characteristics at the downwind ridge.

| | SW wind direction | | NE wind direction | |
|---|---|---|---|---|
| | recirculation | no recirculation | recirculation | no recirculation |
| $\overline{I_{\text{upwind}}}$ [%] | 13.0 | 13.5 | 11.2 | 8.4 |
| $\overline{I_{\text{downwind}}}$ [%] | 20.8 | 16.6 | 13.1 | 9.2 |
| $\overline{\Delta U}$ [%] | -7.1 | 2.5 | 15.5 | 19.3 |
| N [-] | 142 | 118 | 225 | 230 |

are located), or 455 (260) periods to consider for northeasterly (southwesterly) flow. The measurements of 10-minute mean wind speed $U$ (we drop the bar for simplicity) at the downwind ridge are normalized for each period by dividing with the measurements taken at the upstream ridge:

$$\Delta(U) = \frac{(U_{\text{downwind}} - U_{\text{upwind}})}{U_{\text{upwind}}} \cdot 100; \tag{3}$$

5 At both ridges, data from the 100-m sonic anemometers are used. For both wind directions, a decrease in mean wind speed and an increase in turbulence intensity when recirculation is present is found at the downwind ridge (Table 4). For southwesterly wind directions which tended to have larger and more vigorous recirculation zones, these changes are more pronounced. Notice, the mean of $I$ for recirculation periods shows a increase of $I$ at the downwind ridge compared to upwind ridge of 7.8% (1.9%) for southwesterly (northeasterly) wind directions. During periods without recirculation $I$ only increases by 3.1% (0.8%) for
10 southwesterly (northeasterly) wind directions.

## 5 Conclusions

Recirculation zones at two parallel ridges have been analyzed using line-of-sight measurements from six long-range Wind-Scanners and two 100-m measurement masts equipped with sonic anemometers. The data were collected during the Perdigão

2017 measurement campaign in spring and summer 2017 in central Portugal. An method is developed to detect recirculation zones from RHI scans performed at three transects perpendicular to the ridges. Atmospheric stability is characterized using measurements of sonic anemometers and temperature sensors on a 100-m mast on the ridges.

For flow perpendicular to the ridges, recirculation occurs frequently, in 52% of the scans, 55% for southwesterly wind and 49% for northeasterly. The occurrence of recirculation depends on mean wind speed and atmospheric stability. Recirculation is more likely for wind speeds above $8\,\mathrm{m\,s^{-1}}$ (measured 100 m above the ridges) and is less likely during stable atmospheric conditions.

An inter-comparison of the recirculation occurrence per transect revealed significant differences. The northwestern transect shows a 69% probability of recirculation to occur, compared to a 32% probability at the southeastern-most transect, perhaps due to the topographic variations between the transects.

Finally, recirculation affects the wind characteristics which are of importance for wind energy generation at the downwind ridge. During recirculation periods, the mean wind speed is lower at the downwind ridge than when recirculation is not present. Further, turbulence intensity is increased at the downwind ridge. These increases in turbulence intensity were not symmetric: a increase of the mean turbulence intensity of 7.8% (3.1%) is found at the downwind ridge for southwesterly (northeasterly) flow compared to the upwind ridge. These differences can result from differences in the orography: slopes at the southwest ridge are steeper compared to the northeast ridge and the southwest ridge is higher than the northeast ridge. These findings are of importance for wind energy projects in complex terrain. Developers may reduce loads on turbines, and thereby increase turbine lifetimes, by considering the spatial extent of recirculation zones and positioning the turbines accordingly. Because assessment with multiple scanning lidars, as presented in this study, is yet not feasible for commercial projects, efforts should be made to account for recirculation in the flow models used for wind resource assessment.

*Data availability.* All data sets measured during the Perdigão 2017 campaign are publicly available through dedicated web portals of the University of Porto (https://windsp.fe.up.pt/) and NCAR/EOL (http://data.eol.ucar.edu/master_list/?project=PERDIGAO). The lidar dataset measured by DTU is additionally available under the following DOI: https://doi.org/10.11583/DTU.7228544.v1

*Author contributions.* R. Menke performed the analysis and wrote the main body of the manuscript. J. K. Lundquist, J. Mann and N. Vasiljević gave scientific advice. R. Menke, J. Mann and N. Vasiljević designed the lidar part of the field campaign. All authors contributed with critical feedback to this manuscript.

*Competing interests.* The authors declare that they have no conflict of interest.

*Acknowledgements.* We acknowledge the work of everyone involved in the planning and execution of the campaign, in specific we would like to thank Stephan Voß, Julian Hieronimus (ForWind, University of Oldenburg), Per Hansen and Preben Aagaard (DTU Wind Energy) for their help with the installation of the WindScanners and Steven Oncley and Kurt Knudson (National Center for Atmospheric Research) for their help with the installation of the meteorological masts. We are also grateful for the contribution of three WindScanners to the campaign by ForWind. Moreover, only the intensive negotiations of José Carlos Matos, INEGI with local landowners about specific locations for our WindScanners made this research possible. We are grateful to the municipality of Vila Velha de Ródão, landowners who authorized installation of scientific equipment in their properties, the residents of Vale do Cobrão, Foz do Cobrão, Alvaiade, Chão das Servas and local businesses who kindly contributed to the success of the campaign. The space for operational centre was generously provided by Centro Sócio-Cultural e Recreativo de Alvaiade in Vila Velha de Rodão. We thank the Danish Energy Agency for funding through the New European Wind Atlas project (EUDP 14-II). J. K. Lundquist's effort is supported by the U. S. National Science Foundation under grant AGS-1565498.

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
