# Peer review of "Characterization of flow recirculation zones at the Perdigão site using multi-lidar measurements"

_Atmospheric Chemistry and Physics, 2018_

## Referee Comment (RC1) · Anonymous Referee #1 · 29 Sep 2018

This paper reports measurements over a double ridge from a range of instruments (sonic anemometers on 100 m wind masts, Doppler lidars, etc.) made during the Perdigao field campaign in Portugal. The study focuses mostly on the occurrence of flow separation in the lee of the upwind hill, because of the importance for wind-power applications of the decrease of mean wind speed and increase of turbulence intensity that accompanies flow separation (and its effect on the downwind hill). The manuscript represents a substantial contribution to scientific progress, within the scope of ACP, being especially relevant due to the novel and comprehensive data gathered using state-of-the-art equipment. These data are processed and interpreted appropriately, although a number of improvements are possible (see below). The scientific approach, methodology and assumptions seem overall sound, and are described and discussed

in an adequate, clear and balanced way, including allusions to previous work and use of references. Relevant results and conclusions supported by them are presented. The paper is concise, well-structured, clear, and well written. The title reflects the contents of the paper, and the abstract provides a good summary. The quality and number of figures and tables seems appropriate (for the content as it currently stands). The equations, symbols and units are properly defined and used. Although I will suggest some additions that require new calculations and possibly new figures, these should be straightforward to obtain from the existing data, so the paper is likely to require only minor revisions prior to publication.

General comment

Analysis of the collected data could be a bit more comprehensive, specifically regarding conditions under which flow separation is expected. These conditions may be estimated from simple, mostly linear, theories of flow over orography. Even ignoring boundary layer effects (which would complicate the picture considerably), flow separation can be viewed as an outcome of flow deceleration by the orography. Two paradigms may be considered in this problem. For neutral flow, nonlinearity may be quantified by (h/a) (the orography steepness), where h and a are typical height and width for the orography, and the flow perturbation scales as U(h/a), where U is the incoming wind speed. For substantially stratified flow, on the other hand, nonlinearity is quantified by (Nh/U), where N is the Brunt-Vaisala frequency of the incoming wind, and the horizontal flow perturbation scales as (Nh). The authors consider Ri as a relevant parameter, but overlook the sensitivity of the flow to (Nh/U), which would be equally easy to test and is even more basic (since it does not involve vertical derivatives of U). From the results presented in the paper, one gets the impression that neutral flow always causes flow separation (theoretically, this is predicted by (h/a), which is fixed by the orography), but in statically stable conditions the important nonlinearity parameter becomes instead (Nh/U), which might explain the absence of wave breaking detected when Ri>0, i.e. $N^2>0$. With this conceptual framework, the fact that flow separation occurs for high

wind speeds may be explained by the consequent smallness of (Nh/U). The role currently played by Ri, in my view, is primarily distinguishing between unstable or neutral flows, on the one hand, and stable flows, on the other, but it is not obvious that the wind shear effect contained in Ri has much relevance. This is one of the reasons why I suggest that the scaling with (Nh/U) be tested.

Specific comments

Page 1, lines 21-22: "flow acceleration and channeling effects, the formation of lee waves, and flow recirculation (Stull, 2012) which are not captured well by a linearized flow model". The word "well" is essential for this passage not to be grossly inaccurate. Linear models, with a structured atmosphere, are capable of predicting lee waves (Teixeira and Miranda, 2017), and can even give qualitative indications about flow channelling and recirculation (Teixeira, 2017). Linear models are almost the only way to obtain systematic scalings for the flow variables, and that should be recognized more in this passage.

Page 4, line 5: "Range gates". It is not obvious to the reader what these mean exactly. Please add a brief description.

Page 6, lines 4-5: "[in order to calculate the Richardson number] we calculate the difference between the wind speed measured by the 100 m sonic and at the ground level, 0 m, where the wind speed is assumed to be zero". I assume that in this calculation and that of the vertical potential temperature gradient used to evaluate Ri the authors adopt linear interpolation in Eq. (1) (these details are not specified). The range of heights over which this calculation is performed is most likely in the surface layer. Since the profiles of both wind speed and potential temperature in that layer are logarithmic to a first approximation, it might be more accurate to determine Ri based on the logarithmic finite-difference approximation described by Arya (2001), Eq. (11.22). If this is not appropriate, please justify why.

Figures 5 and 6: following my suggestions in the General Comment above, it would be

probably useful to produce figures similar to these, but where Ri on the horizontal axis is replaced by (Nh/U), or U/(Nh). This should give some additional physical insight into the flow behaviour.

Page 9, lines 2-3: "Recirculation is more likely to occur during unstable or neutral atmospheric conditions (Ri<0) than for stable conditions (Ri>0) for both SW and NE winds". The interpretation presented in my General Comment, along with plotting the data as a function of (Nh/U), might be able to shed some light as to why this happens. In terms of the behaviour with Ri, what appears to matter (see below) is the sign of $N^2$, more than the detailed value of Ri. Perhaps this could be recognized and discussed.

Page 9, lines 13-18: Flow separation is discussed with relation to the height and steepness of the ridges. It is also noted that the southwest slope of the soutwest ridge is steeper than the northeast slope of the northeast ridge. However, for flow separation in southwest flow what should be most important is the steepness of the northeast slope of the southwest ridge (because separation occurs downwind of obstacles) and for northeast flow the steepness of the southwest slope of the northeast ridge. The authors should check whether the steepnesses of these slopes are consistent with this physical interpretation of the results.

Figure 8: the reverse flow speed is presented as a function of the upstream flow speed. As is consistent with my General Comment above, this tests a scaling for neutral flow, where the velocity perturbation scales as $u \sim U(h/a)$. This scaling is likely to be applicable, in Fig. 8, to the points that have Ri<0 (i.e. $N^2<0$), because they will generate no orographic gravity waves. For points that have Ri>0 (i.e. $N^2>0$), orographic gravity waves will occur, and the corresponding stratified scaling may apply, namely $u \sim (Nh)$. So, it would be interesting to produce a figure similar to Fig. 8, but with (Nh) in the horizontal axis. Perhaps a better collapse will be obtained for points with $N^2$ (although there are definitely many other processes going on, most prominently boundary layer effects). Even if this scaling with (Nh) does not work very well (for example, due to an insufficiently strong stratification), it is interesting to compare the two scalings.

**References**

Teixeira, M. A. C. (2017) Diagnosing lee wave rotor onset using a linear model including a boundary layer. Atmosphere, vol. 8, 5.

Teixeira, M. A. C. and Miranda, P. M. A. (2017) Drag associated with 3D trapped lee waves over an axisymmetric obstacle in two-layer atmospheres. Quarterly Journal of the Royal Meteorological Society, vol. 143, 3244-3258.
* * *

---

## Referee Comment (RC2) · Anonymous Referee #2 · 6 Oct 2018

This manuscript uses Doppler lidar observations collected over the complex terrain of Perdigao to provide a qualitative description of recirculation on the lee side of the ridges. Recirculation was mostly observed under periods of neutral and unstable stratification during measurement campaign. Recirculation was also observed during stable conditions, albeit less frequently. Authors has made an effort to analyze the occurrence of recirculation along three transects and during southwesterly and northwesterly winds. The results of this paper could be useful for wind turbine siting in complex terrain, as well as to assess the performance of wind solvers qualitatively. Therefore, I support the publication of this manuscript after a revision that addresses the following issues.

1) Based on the content of the manuscript and the extent of the analysis, "Qualitative

characterization of flow recirculation zones in complex terrain using multi-lidar mea-surements" would be a more adequate title for the investigation presented.

2) Vertical profiles of wind speed and direction during certain periods are needed to enable computational researchers to simulate the problem. These profiles should be extracted from multiple locations such from ridgetops and inside the recirculation zone and provided as new figures along with corresponding Richardson number.

3) Page 1 Line 15. The introduction to wind turbine siting is out of date (e.g. the cited reference is from 1989) and does not reflect the latest best practices. The discussion needs to be updated to reflect the current state of the art in this area.

4) Page 2 Line 5: Change "characterized" to "identified". (i.e. flow recirculation can be identified . . .)

5) Page 2 Line 5: Regarding authors' discussion of the Kutter et al (2017). There is nothing unexpected about recirculation being "prevalent" during neutral or unstable conditions. If the hill or any obstruction is steep enough, the flow is expected to recir-culate in the wake under those conditions. Therefore, instead of saying "Kutter et al. "find" recirculation prevalent during . . ." authors could say: "for instance, recirculation was prevalent during neutral and unstable atmospheric conditions during the observa-tional study of Kutter et al.

6) Page 2 Line 10: Recirculation was intermittent in the Askervein experiment and the focus of Askervein was not to study recirculating flows. It would be better to refer to those studies as complex terrain studies as opposed to recirculation studies.

7) There is no mention of the Bolund Hill experiment and related studies. Introduction section need to review those recent efforts since the Askervein case for completeness.

8) Figure 1: Please provide the dominant wind direction observed during the measure-ments and consider using a different marker for the wind turbine and refer to the Met Tower as Mast in the legend to be consistent with the text.

9) Page 4, the paragraph section 3: Figure 1 should be redrawn to convey the information given in this paragraph about wind directions (i.e. the dominant wind directions should be overlayed on the figure.

10) Page 5: Include a subscript G to emphasize gradient Richardson number in Equation 1

11) Figure 2: Make it larger for researchers who may need to digitize it. Create labels for SW Ridge and NE Ridge on the graph for sake of convenience for the readers.

12) Figure 4: Mark SW and NE ridges with labels on the terrain. Provide the gradient Richardson number for this 10 min period.

13) Figure 5: Increase the intervals in the x-axis so that the reader can approximately extract the Ri values without needing to digitize the graph. The bin width information in the caption is not helpful.

14) Figure 5: It would be more useful to present Figure 5 per transect as done in Figure 6, but for the Ri number.

15) Page 8 line 5: The features that are mentioned might play a role in the non-existence of recirculation, but the word "infer" is too strong in my opinion. Without a more detailed analysis, these features are suspects at best and insufficient to infer any flow behavior.

16) Page 9 line 5: Similar concern as in 5). The manuscript conveys the occurrence of recirculation in neutral and unstable conditions as if it is an unexpected feature. Recirculation under those conditions for steep geometry or terrain are expected without any surprise. The more interesting finding would be recirculation under stable conditions, which is much more interesting. The discussion can be revised to describe observations and results that are expected and do not qualify as "findings"

17) Authors can be more precise in their use of the term "stable conditions. Stable conditions need to be categorized as weakly, moderately and strongly stable based on

the Ri number at hand, and authors can then compare against other studies that has similar conditions under that categorization.

18) Authors are only relying on a generic categorization of stable conditions to explain the existence or non-existance of recirculation. Flow separation is highly dependent on the geometry. The current discussion fails to explain why recirculation exist or does not exist under stable conditions.

19) Conclusions: Provide the height for the 8 m/s wind speed.

20) Conclusions: Typo in the last sentence. "Should be made".
* * *

---

## Referee Comment (RC3) · Anonymous Referee #3 · 2 Nov 2018

General considerations

In this paper, the authors use a wonderful data set (from the Perdigão-2017 field campaign) to describe the 'characteristics of flow recirculation zones in complex terrain'. Overall, the paper is quite well written and the provided material is useful to serve the purpose of the paper. However, some of the 'ingredients' of what is called an 'algorithm' to detect recirculation zones and some of the analysis tools need some clarification and reasoning (see the 'minor comments').

My major concern with this paper lies in the embedding of the obtained results in previous knowledge. The authors have decided to 'wipe this away' with a single comment ('. . . which are not well captured by a linear flow model', p1, l. 22) and therefore analyze the data with respect to each variable separately. First of all, linear theory is not all we

know about flow behavior over topography, but more importantly, this decision leads to a 'characterization' that is entirely specific to the Perdigao site: all the given 'numbers' (e.g., in the conclusions: 'recirculation is more likely for wind speed above 8 ms-1', p13, l.9) are not suitable to be transferred to any other site. In the end, the authors then (have to) conclude 'Because assessment with multiple scanning lidars, as presented in this study, is yet not feasible for commercial projects, efforts should be [made] to account for recirculation in the flow models used for wind resource assessment'. While this is certainly a valid conclusion, in my view, additional generality could (easily) be obtained when at least discussing (using) some of the principles arising from previous knowledge. In this sense, the present 'descriptive' approach appears to be a missed opportunity. Clearly, a review cannot require from the authors to change their strategy of analysis – but I think the least that should be done is to comprehensively summarize the previous knowledge and discuss the 'departures' of the present site from the conditions, for which we have some theoretical understanding (major comment below). If the authors decide to keep their approach of analysis with respect to dimensional variables, the title should be changed (into something like 'Characterization of flow recirculation zones at the Perdigao site using multi-lidar measurements')

Major comment

Occurrence of recirculation behind a two-dimensional or three-dimensional obstacle has been widely investigated for stratified flow. Baines (1995) provides an excellent overview – see especially his Chapter 5. The characteristics of the mean flow behavior (over topography) are largely determined by the relative importance of stratification and advection in combination with the 'obstacles' dimensions (half-width, height) leading to an assessment of the flow characteristics in terms of non-dimensional parameters, Froude number (using the half-width of the ridge) or non-dimensional hill height (using the height).

At least for the stable cases (which are not abundant at the present location), therefore, the authors' findings could be compared to previous knowledge with theoretical

foundation. For example, Fig. 5.8 in Baines (1995) summarizes the occurrence of flow separation in a non-dimensional framework – and it would be extremely interesting to learn to what degree the re-circulation occurrence of the different cross-sections at the present site correspond to those (mostly ideal) results. Certainly, on the basis of this previous knowledge, it would be much more conclusive to analyze the present data in a non-dimensional framework, rather than producing 'thresholds' for the mean wind and stability separately – and finding of course results that are surely consistent with this (larger mean wind speed favors separation, stronger stability hinders it) and then speculating that 'These variations of recirculation occurrence may be related to the transects' elevation profiles within the valley', [p8, l.1]).

One of the relevant (and quite new) findings of the present study is certainly that re-circulation preferably occurs under unstable conditions (at the present site). This of course makes it more difficult to compare to previous knowledge for stratified flows. Even if the authors 'rule out' the value of linear theory (p1, l. 22), the Perdigao site is sufficiently ideal (and slopes may be steeper than desired for linear theory, but certainly not overly steep) that at least the consistency (in the trends) of the present results with the expectations from linear theory could be discussed. In fact, should be. The textbook of Kaimal and Finnigan (1994, their chapter 5) is an excellent source to start with – and Belcher and Hunt (1998) give a comprehensive overview of all the relevant resources. Again, stability is characterized in a non-dimensional framework using the terrain geometry (in this case the 'inner-layer depth', which could be determined from the present data) thus making the results more generally applicable.

The strongest 'departure' from applicability of previous knowledge is the 'double ridge' problem, i.e. the fact that at the Perdigao site not only one ridge is present but two – and those in a short enough distance so that they potentially influence the flow at each others' location. There is less systematic knowledge available for this flow type with respect to recirculation zones – except for the quite specific case of rotor formation (which, of course, is also some sort of 're-circulation zone', but not immediately

downstream of the ridge. See Grubisic et al. (2008) for some detail). Again, for the stable case, some information can be found in Baines' book (but clearly less, and less theoretically founded). I think, this aspect may indeed be used in the discussion of the present results in view of potential departures from expectations for a single ridge.

Minor comments

P2/l. 28 Section 4 (not section)

Fig 1 caption: 'Table 1' (when using in conjunction with a number, please capitalize: Tab. 1,

Fig. 2 etc.). Throughout - many occurrences.

P2/l.10 many studies...: there are quite some others, e.g. discussed in Kaimal and Finnigan (1994).

P2/l.18 ...the orography of the Perdigao site is more complex...: likely, the Perdiagao site is as close to an ideal 'two-dimensional ridge' [more precisely, a valley between two two-dimensional ridges] as Askervein is to an ideal 3d hill. What is different is the dimensionality of the obstacle(s) and the slopes.

P3/l.7 SW and NE ridge, respectively.

P4, l.1 ....for the exact directions: directions are not really provided in this table (of course, one could determine them by additionally using Fig. 1...).

P4/l.18 ...due TO missing.... However, until now we have only learned that two 100 m masts were used with sonics (p.3, l. 1) – now, all of a sudden, we have temperature measurements (which failed). Given the well-known problems with (absolute) accuracy of sonic temperature measurements, I do not hope that the authors have used the sonic temperatures to calculate Ri (also, it is hard to imagine that the sonic produces wind but no temperature...). It seems that the list of instruments (on p. 3) should be completed.

Eq (1) In the equation, the authors write 'T', and after the equation they explain 'T

overbar' (i.e., mean temperature). This must be consistent. However, the buoyancy term in the definition of the gradient Richardson number, in fact, should be defined with the potential temperature (not T), see, e.g., Stull (2012). While usually in Surface Layer micrometeorology, this is not really relevant, over the height of 100 m, this makes a difference.

P7/ l. 8 'In overlap regions only measurements of the lidar further downwind are used'. I am not sure whether I understand this. What I understand then looking at Fig. 4a, the 'lidar further downwind' for the entire overlap region (which is the purple part) is the blue one, right? So, the second lidar is almost obsolete (only the really small part of the 'rim' is from this instrument...). This needs some reasoning. Also, looking at Fig. 4 suggests that the two lidars did not scan the same range (opening angle of the RHI). This should i) be mentioned where the scan strategy is introduced and ii) be motivated.

P7/l.11 '...we do not attempt...': this seems to be at odds with l.7 (Cartesian coordinate system ....with the abscissa ). The authors certainly need some rotation to do this, right? Does it mean that only one elevation angle for each RHI is used? I cannot see why this should be any easier to handle (and I cannot see what 'complex terrain' has to do with this). Can the authors please explain this procedure in some more consistency?

P9/l.15 . . . is in general more . . .

Fig 7 Unfortunately, the color coding in this figure is the opposite to that of Fig. 4.

Fig. 8 'reverse flow speed': this should be specified (median, average over the recirculation zone, etc.).

P11, l.11 'also influences [the] flow ...': given the purely descriptive approach taken in this paper, in a case study one cannot conclude from one case that 'the leeward recirculation zone and wake ..... decrease the mean wind speed...'. The wording has to be much more cautious (something like 'for this case with a strong recirculation

zone we observe a reduction in mean wind speed...'). Clearly, other cases with similar conditions but no recirculation zone would be needed to allow for a conclusion like 'due to the recirculation zone wind speed is smaller and turbulence intensity larger'.

P12/eq (2) I am not convinced that the ratio of the median values is an extremely useful measure to demonstrate the slow-down (increase in turbulence intensity). This would be a (statistically) appropriate measure, if wind speed were normally distributed (but often it is not). Is there any reason not to use a proper 'delta' (U_ds - U_us)/U_us ('ds'=downstream, 'us'=upstream), and average over all cases? If the sign of this measure were significantly different from zero (and significance can be tested...) – and even 'more negative' for recirculation occurrence than for no occurrence, this would be a strong indication that there is a reduction in mean speed (increase in intensity) due to the recirculation zone.

Tab. 4 If the 'delta's are defined as in eq (2), the given information is not in % (if the median ratio for 'recirc' is 0.42, say, and that for 'no recirc' is 0.50, the indicated difference amounts to -0.08 – and not even multiplying with 100 makes this to be an 8.2% reduction....). Definition of those 'delta's and their use should be made clear and the wording adjusted.

P13/l.4 'Algorithms are developed...': In fact, it is only one – and it is not really an algorithm, but rather a straight forward ad hoc procedure.

P13/l.21 should be made

References:

Baines PG: 1995, Topographic effects in stratified flows, Cambridge University Press, Cambridge, 482 pp.

Belcher SE and Hunt JRC: 1998, Turbulent flows over hills and waves, Ann Rev Fluid Mech, 30, 507-538.

Grubišic V, Doyle JD, Kuettner, J et al.: 2008, The terrain-induced rotor experiment,

Bull American Meteorol Soc, 89, 1513–1533.

Kaimal JC and Finnigan JJ: 1994, Atmospheric Boundary Layer Flows, Oxford University Press, New York, 289 pp.

---

## Author Comment (AC1) · 31 Jan 2019

**Response to reviewer 1**

**Dear Anonymous Reviewer 1,**

we highly appreciate your feedback. It helped us to improve the manuscript. Below we comment on your suggestions in detail.

This paper reports measurements over a double ridge from a range of instruments (sonic anemometers on 100 m wind masts, Doppler lidars, etc.) made during the Perdigao field campaign in Portugal. The study focuses mostly on the occurrence of flow separation in the lee of the upwind hill, because of the importance for wind-power applications of the decrease of mean wind speed and increase of turbulence intensity that accompanies flow separation (and its effect on the downwind hill). The manuscript represents a substantial contribution to scientific progress, within the scope of ACP, being especially relevant due to the novel and comprehensive data gathered using state-of-the-art equipment. These data are processed and interpreted appropriately, although a number of improvements are possible (see below). The scientific approach, methodology and assumptions seem overall sound, and are described and discussed in an adequate, clear and balanced way, including allusions to previous work and use of references. Relevant results and conclusions supported by them are presented. The paper is concise, well-structured, clear, and well written. The title reflects the contents of the paper, and the abstract provides a good summary. The quality and number of figures and tables seems appropriate (for the content as it currently stands). The equations, symbols and units are properly defined and used. Although I will suggest some additions that require new calculations and possibly new figures, these should be straightforward to obtain from the existing data, so the paper is likely to require only minor revisions prior to publication.

**General comment**

Analysis of the collected data could be a bit more comprehensive, specifically regarding conditions under which flow separation is expected. These conditions may be estimated from simple, mostly linear, theories of flow over orography. Even ignoring boundary layer effects (which would complicate the picture considerably), flow separation can be viewed as an outcome of flow deceleration by the orography. Two paradigms may be considered in this problem. For neutral flow, nonlinearity may be quantified by (h/a)(the orography steepness), where h and a are typical height and width for the orography, and the flow perturbation scales as U(h/a), where U is the incoming wind speed. For substantially stratified flow, on the other hand, nonlinearity is quantified by (Nh/U), where N is the Brunt-Vaisala frequency of the incoming wind, and the horizontal flow perturbation scales as (Nh). The authors consider Ri as a relevant parameter, but overlook the sensitivity of the flow to (Nh/U), which would be equally easy to test and is even more basic (since it does not involve vertical derivatives of U). From the results presented in the paper, one gets the impression that neutral flow always causes flow separation (theoretically, this is predicted by (h/a), which is fixed by the orography), but in statically stable conditions the important nonlinearity parameter becomes instead (Nh/U), which might explain the absence of wave breaking detected when Ri>0, i.e.  $N^2>0$ . With this conceptual framework, the fact that flow separation occurs for high wind speeds may be explained by the consequent smallness of (Nh/U). The role currently played by *Ri, in my view, is primarily distinguishing between unstable or neutral flows, on the one hand, and stable*

flows, on the other, but it is not obvious that the wind shear effect contained in Ri has much relevance. This is one of the reasons why I suggest that the scaling with (Nh/U) be tested.

Thank you for outlining this theoretical background. We agree that we did not cover it well in the manuscript and included a new subsection (4.1.4) discussing our measurements in contrast to the results and findings of past experiments. More precisely, we present our result in the same non-dimensional framework as used in Baines (1995) Fig. 5.8 of h/A over Nh/U.

**Specific comments**

Page 1, lines 21-22: "flow acceleration and channeling effects, the formation of lee waves, and flow recirculation (Stull, 2012) which are not captured well by a linearized flow model". The word "well" is essential for this passage not to be grossly inaccurate. Linear models, with a structured atmosphere, are capable of predicting lee waves (Teixeira and Miranda, 2017), and can even give qualitative indications about flow channelling and recirculation (Teixeira, 2017). Linear models are almost the only way to obtain systematic scalings for the flow variables, and that should be recognized more in this passage.

Thanks for this comment. We meant to refer with the expression "*linearized flow model*" to software that is used in the wind farm planning process. We updated the manuscript, it reads now as follows: "A vast number of flow phenomena occur at sites with complex geometry (Rotach and Zardi, 2007), such as flow acceleration and channeling effects, the formation of lee waves, and flow recirculation (Stull, 2012) which are not captured well by frequently applied computer models (e.g. the Wind Atlas Analysis and Application Program, WAsP) in the wind farm planning process."

Page 4, line 5: "Range gates". It is not obvious to the reader what these mean exactly. Please add a brief description.

A description of the term has been added to the manuscript. The manuscript reads now as follows: "Range gates (time intervals used for determining the wind speed from the backscattered light. These time intervals corresponds to spatial intervals along the line-of-sight (LOS) for which the LOS wind speed is evaluated. They translate into a weighting function along the LOS which in this case has a full-width half-maximum of approximately 30 m.) were placed every 15 m, starting at a range of 100 m extending to 3000 m."

Page 6, lines 4-5: "[in order to calculate the Richardson number] we calculate the difference between the wind speed measured by the 100 m sonic and at the ground level, 0 m, where the wind speed is assumed to be zero". I assume that in this calculation and that of the vertical potential temperature gradient used to evaluate Ri the authors adopt linear interpolation in Eq. (1) (these details are not specified). The range of heights over which this calculation is performed is most likely in the surface layer. Since the profiles of both wind speed and potential temperature in that layer are logarithmic to a first approximation, it might be more accurate to determine Ri based on the logarithmic finite-difference approximation described by Arya (2001), Eq. (11.22). If this is not appropriate, please justify why.

In general, the estimation of the atmospheric stability is associated with high uncertainties, especially in complex terrain that we are investigating in a separate study in more detail. While a logarithmic approximation of the potential temperature profiles may be applicable under special conditions, our investigation of these profiles from the 100-m towers at Perdigao suggests that assumption is not reliable at this site (compare Stull (2003) Fig. 5.17) and we found a linear interpolation as used here may be more accurate. In our initial investigation, which included a comparison of several approaches for calculating the Richardson number, the Brunt-Vaisala frequency and Obukhov length, we found the Richardson number as a most reliable parameter to estimate the stability. (By "reliable", we mean provided stability assessments consistent with time-of-day and did not lead to abrupt unreasonable transitions) Further, for the wind speed, a logarithmic approximation of the profile cannot be used since we decided to assume 0 m/s at 0 m height to minimize the effect of terrain-induced flow. An similar approach like ours has been used by Burns et al. (2011) and the reference has been added to the manuscript.

Figures 5 and 6: following my suggestions in the General Comment above, it would be probably useful to produce figures similar to these, but where Ri on the horizontal axis is replaced by (Nh/U), or U/(Nh). This should give some additional physical insight into the flow behaviour.

Thank you for the helpful suggestion. We have now included in the new section 4.1.4 a figure in which we present our observation under stratified conditions in the non-dimensional framework of h/A over Nh/U (see Figure 1 below).

Figure 1: Observational periods with recirculation (colored markers) and without recirculation (gray cross marks) under stratified conditions as a function of mean lee-side slope (h/A) and Nh/u. The round markers refer to recirculation at the southwest ridge and the square markers to northeast ridge. The colored diamond markers show the mean value of Nh/u per transect during periods with recirculation and the gray markers the mean for periods without recirculation. The solid black line shows where NA/u =  $\pi$ .

Page 9, lines 2-3: "Recirculation is more likely to occur during unstable or neutral atmospheric conditions (Ri<0) than for stable conditions (Ri>0) for both SW and NE winds". The interpretation presented in my General Comment, along with plotting the data as a function of (Nh/U), might be able to

shed some light as to why this happens. In terms of the behaviour with Ri, what appears to matter (see below) is the sign of  $N^2$ , more than the detailed value of Ri. Perhaps this could be recognized and discussed.

Thanks for this comment. As mentioned above, we included section 4.1.4 which included a discussion of the mechanisms that are present under stratified conditions including a discussion of Nh/U..

Page 9, lines 13-18: Flow separation is discussed with relation to the height and steepness of the ridges. It is also noted that the southwest slope of the southwest ridge is steeper than the northeast slope of the northeast ridge. However, for flow separation in southwest flow what should be most important is the steepness of the northeast slope of the southwest ridge (because separation occurs downwind of obstacles) and for northeast flow the steepness of the southwest slope of the northeast ridge. The authors should check whether the steepnesses of these slopes are consistent with this physical interpretation of the results.

We appreciate this comment and updated the manuscript accordingly. The passage reads now as follows: "A possible explanation for this difference could be that the northeast face of the southwest ridge (the downwind face in southwesterly flow) has steep escarpments close to the ridge top and the average slope is slightly steeper compared to the southwest face of the northeast ridge (the downwind face in northeasterly flow). Both the higher steepness and the escarpments make flow separation more likely."

Figure 8: the reverse flow speed is presented as a function of the upstream flow speed. As is consistent with my General Comment above, this tests a scaling for neutral flow, where the velocity perturbation scales as  $u \sim U(h/a)$ . This scaling is likely to be applicable, in Fig. 8, to the points that have Ri<0 (i.e.  $N^2 < 0$ ), because they will generate no orographic gravity waves. For points that have Ri>0 (i.e.  $N^2 > 0$ ), orographic gravity waves will occur, and the corresponding stratified scaling may apply, namely  $u \sim (N h)$ . So, it would be interesting to produce a figure similar to Fig. 8, but with (Nh) in the horizontal axis. Perhaps a better collapse will be obtained for points with  $N^2$  (although there are definitely many other processes going on, most prominently boundary layer effects). Even if this scaling with (Nh) does not work very well (for example, due to an insufficiently strong stratification), it is interesting to compare the two scalings.

Thanks for this comment. We tested the scaling with N2 and it is definitely also interesting but doesn't give necessarily better insights than the use of the Richardson number. However, we decided to change Figure 8 to a dimensionless representation of reverse wind speed divided by the inflow wind speed over the Richardson number (Figure 2). Additionally, we generated Figure 3 that shows a comparison of the two scalings. This comparison of N2 and Richardson number shows that both metrics agree well in the general classification of the different stability regimes.

Figure 2: Maximum reverse flow speed divided by the inflow wind speed over the Richardson number. The color scaling shows the wind speed at the mast upstream.

Figure 3: Comparison of  $N^2$  and the Richardson number for wind speeds over 3 m/s. The color scaling shows the wind speed at the mast upstream.

Burns, S.P., Sun, J., Lenschow, D.H., Oncley, S.P., Stephens, B.B., Yi, C., Anderson, D.E., Hu, J. and Monson, R.K., 2011. Atmospheric stability effects on wind fields and scalar mixing within and just above a subalpine forest in sloping terrain. *Boundary-layer meteorology*, *138*(2), pp.231-262.

**Response to reviewer 2**

Dear Anonymous Reviewer 2,

thank you for your critical feedback which helped us to improve this manuscript. Below we answer the specific comments in detail.

This manuscript uses Doppler lidar observations collected over the complex terrain of Perdigao to provide a qualitative description of recirculation on the lee side of the ridges. Recirculation was mostly observed under periods of neutral and unstable stratification during measurement campaign. Recirculation was also observed during stable conditions, albeit less frequently. Authors has made an effort to analyze the occurrence of recirculation along three transects and during southwesterly and northwesterly winds. The results of this paper could be useful for wind turbine siting in complex terrain, as well as to assess the performance of wind solvers qualitatively. Therefore, I support the publication of this manuscript after a revision that addresses the following issues.

1) Based on the content of the manuscript and the extent of the analysis, "Qualitative characterization of flow recirculation zones in complex terrain using multi-lidar measurements" would be a more adequate title for the investigation presented. We appreciate this suggestion.

The request for the modification of the title was also raised by another anonymous reviewer. Accordingly, we changed the title of our manuscript to :

"Characterization of flow recirculation zones at the Perdigão site using multi-lidar measurements" in order to reflect that this study was focused on the findings at the Perdigão site.

2) Vertical profiles of wind speed and direction during certain periods are needed to enable computational researchers to simulate the problem. These profiles should be extracted from multiple locations such from ridgetops and inside the recirculation zone and provided as new figures along with corresponding Richardson number.

The objective of this manuscript is not to provide model initiation data. All data sets measured during the Perdigão 2017 campaign are publicly available through dedicated Web portals of the University of Porto (https://windsp.fe.up.pt/) and NCAR/EOL (http://data.eol.ucar.edu/master\_list/?project =PERDIGAO). Additionally, the DTU lidar measurement data is available under the following DOI: https://doi.org/10.11583/DTU.7228544.v1. We included the information on the data availability in the manuscript. This gives the opportunity to modelers to select initiation data to their needs, which is best suited for their specific application. We and other participants of the campaign are also very open to react on individual request to help with the selection of appropriate measurements.

3) Page 1 Line 15. The introduction to wind turbine siting is out of date (e.g. the cited reference is from 1989) and does not reflect the latest best practices. The discussion needs to be updated to reflect the current state of the art in this area.

The cited reference Troen and Petersen (1989) points to the European Wind Atlas that presents a methodology for wind turbine siting which is today except for small modification still widely used for site assessments. We decided to keep the reference to this work but updated the introduction with information on recent field campaigns (Lange et al., 2016) and modelling approaches (Silva Lopes et al., 2007, Chow and Street, 2009) with the aim to improve site assessments for the wind energy deployments.

*4) Page 2 Line 5: Change "characterized" to "identified". (i.e. flow recirculation can be identified . . .)* Thanks for this correction.

We assume that this comment refers page 2 line 3 and replaced "characterized" with "identified" in the manuscript.

5) Page 2 Line 5: Regarding authors' discussion of the Kutter et al (2017). There is nothing unexpected about recirculation being "prevalent" during neutral or unstable conditions. If the hill or any obstruction is steep enough, the flow is expected to recirculate in the wake under those conditions. Therefore, instead of saying "Kutter et al. "find" recirculation prevalent during . . ." authors could say: "for instance, recirculation was prevalent during neutral and unstable atmospheric conditions during the observational study of Kutter et al.

We agree with the reviewer and changed the passage on page 2 line 8-9 as suggested.

6) Page 2 Line 10: Recirculation was intermittent in the Askervein experiment and the focus of Askervein was not to study recirculating flows. It would be better to refer to those studies as complex terrain studies as opposed to recirculation studies.

We agree that the main aspect of the Askervein was not to only measure recirculation. The passage has been updated.

7) There is no mention of the Bolund Hill experiment and related studies. Introduction section need to review those recent efforts since the Askervein case for completeness. The introduction has been updated and includes now the Bolund hill experiment.

8) Figure 1: Please provide the dominant wind direction observed during the measurements and consider using a different marker for the wind turbine and refer to the Met Tower as Mast in the legend to be consistent with the text.

A wind rose showing the wind distribution measured at the northeast ridge has been added to figure 1. Moreover, the label and marker for the met mast have been changed.

9) Page 4, the paragraph section 3: Figure 1 should be redrawn to convey the information given in this paragraph about wind directions (i.e. the dominant wind directions should be overlayed on the figure. A wind rose showing the wind distribution measured at the northeast ridge has been added to figure 1.

10) Page 5: Include a subscript G to emphasize gradient Richardson number in Equation 1 The subscript has been added. 11) Figure 2: Make it larger for researchers who may need to digitize it. Create labels for SW Ridge and NE Ridge on the graph for sake of convenience for the readers.

Thanks for recommending the use of labels for the two ridges, we included them in the figure 2. The figure is provided in a reasonable resolution, it can be easily scaled up on any computer for the purpose of digitization. Additionally, we included the coordinates of each transect as CSV files in the supplementary material. The terrain data is also part of the openly available datasets shared by the above mentioned institutions.

12) Figure 4: Mark SW and NE ridges with labels on the terrain. Provide the gradient Richardson number for this 10 min period.

Labels for the ridges also have been added for this figure and information about the gradient Richardson number is now provided in the caption.

13) Figure 5: Increase the intervals in the x-axis so that the reader can approximately extract the Ri values without needing to digitize the graph. The bin width information in the caption is not helpful. We believe that the current number of bins presents the distribution well. We increased the size of the figure for better readability.

14) Figure 5: It would be more useful to present Figure 5 per transect as done in Figure 6, but for the Ri number.

We tested to present figure 5 also per wind direction and transect. The separation of the distributions into different transect did not add any additional insights. Accordingly, we decided to retain the current approach in presenting the results.

15) Page 8 line 5: The features that are mentioned might play a role in the non- existence of recirculation, but the word "infer" is too strong in my opinion. Without a more detailed analysis, these features are suspects at best and insufficient to infer any flow behavior.

We agree that the word "infer" might be too strong in this context, it has been replaced with "assume".

16) Page 9 line 5: Similar concern as in 5). The manuscript conveys the occurrence of recirculation in neutral and unstable conditions as if it is an unexpected feature. Re- circulation under those conditions for steep geometry or terrain are expected without any surprise. The more interesting finding would be recirculation under stable conditions, which is much more interesting. The discussion can be revised to describe observations and results that are expected and do not qualify as "findings"

As suggested we made clear that flow recirculation can be expected for neutral and unstable conditions. The passage reads now as follows: "*Recirculation is more likely to occur during unstable or neutral atmospheric conditions* ( $Ri \le 0$ ) than for stable conditions (Ri > 0) for both SW and NE winds (Figure 5), as expect for flow over steep geometries such as the Perdigão ridges."

17) Authors can be more precise in their use of the term "stable conditions. Stable conditions need to be categorized as weakly, moderately and strongly stable based on the Ri number at hand, and authors can then compare against other studies that has similar conditions under that categorization.

We are not aware of any other studies that present a categorization of stable conditions based the Richardson number in complex terrain. In general, a description of how the atmospheric stability can be described in complex terrain is missing. Existing methods to describe the atmospheric stability based on the Richardson number are validated for flat terrain and it is not straightforward to adopt a fine categorization for measurements in complex terrain. However, we included subsection 4.1.4 analysing our findings under stably stratified conditions in an non-dimensional framework which gives additional insights.

18) Authors are only relying on a generic categorization of stable conditions to explain the existence or non-existance of recirculation. Flow separation is highly dependent on the geometry. The current discussion fails to explain why recirculation exist or does not exist under stable conditions.

Subsection 4.1.4 which has been added to the reviewed version of the manuscript gives additional insights into the recirculation occurrences under stable conditions.

*19)* Conclusions: Provide the height for the 8 m/s wind speed. We included the measurement height in the manuscript.

*20) Conclusions: Typo in the last sentence. "Should be made".* Corrected.

**Response to reviewer 3**

Dear Anonymous Reviewer 3,

thank you for your extensive and constructive comments. Your feedback is highly appreciated. We comment on the suggestions in detail below.

**General considerations**

In this paper, the authors use a wonderful data set (from the Perdigão-2017 field campaign) to describe the 'characteristics of flow recirculation zones in complex terrain'. Overall, the paper is quite well written and the provided material is useful to serve the purpose of the paper. However, some of the 'ingredients' of what is called an 'algorithm' to detect recirculation zones and some of the analysis tools need some clarification and reasoning (see the 'minor comments').

My major concern with this paper lies in the embedding of the obtained results in previous knowledge. The authors have decided to 'wipe this away' with a single comment ('... which are not well captured by a linear flow model', p1, l. 22) and therefore analyze the data with respect to each variable separately. First of all, linear theory is not all we know about flow behavior over topography, but more importantly, this decision leads to a 'characterization' that is entirely specific to the Perdigao site: all the given 'numbers' (e.g., in the conclusions: 'recirculation is more likely for wind speed above 8 ms-1', p13, l.9) are not suitable to be transferred to any other site. In the end, the authors then (have to) conclude Because assessment with multiple scanning lidars, as presented in this study, is yet not feasible for commercial projects, efforts should be [made] to account for recirculation in the flow models used for wind resource assessment'. While this is certainly a valid conclusion, in my view, additional generality could (easily) be obtained when at least discussing (using) some of the principles arising from previous knowledge. In this sense, the present 'descriptive' approach appears to be a missed opportunity. Clearly, a review cannot require from the authors to change their strategy of analysis – but I think the least that should be done is to comprehensively summarize the previous knowledge and discuss the 'departures' of the present site from the conditions, for which we have some theoretical understanding (major comment below). If the authors decide to keep their approach of analysis with respect to dimensional variables, the title should be changed (into something like 'Characterization of flow recirculation zones at the Perdigao site using multi-lidar measurements')

**Major comment**

Occurrence of recirculation behind a two-dimensional or three-dimensional obstacle has been widely investigated for stratified flow. Baines (1995) provides an excellent overview – see especially his Chapter 5. The characteristics of the mean flow behavior (over topography) are largely determined by the relative importance of stratification and advection in combination with the 'obstacles' dimensions (half-width, height) leading to an assessment of the flow characteristics in terms of non-dimensional parameters, Froude number (using the half-width of the ridge) or non-dimensional hill height (using the height). At least for the stable cases (which are not abundant at the present location), therefore, the authors' findings could be compared to provide incovaled a with theoretical foundation.

findings could be compared to previous knowledge with theoretical foundation. For example, Fig. 5.8 in Baines (1995) summarizes the occurrence of flow separation in a non-dimensional framework – and it would be extremely interesting to learn to what degree the re-circulation occurrence of the different

cross-sections at the present site correspond to those (mostly ideal) results. Certainly, on the basis of this previous knowledge, it would be much more conclusive to analyze the present data in a non-dimensional framework, rather than producing 'thresholds' for the mean wind and stability separately – and finding of course results that are surely consistent with this (larger mean wind speed favors separation, stronger stability hinders it) and then speculating that 'These var